# Biologic Therapies in Pediatric Asthma

**DOI:** 10.3390/jpm12060999

**Published:** 2022-06-18

**Authors:** Evanthia P. Perikleous, Paschalis Steiropoulos, Evangelia Nena, Emmanouil Paraskakis

**Affiliations:** 1Medical School, Democritus University of Thrace, 68100 Alexandroupoli, Greece; eviperikleous@hotmail.com; 2Department of Pneumonology, University General Hospital of Alexandroupolis, Democritus University of Thrace, 68100 Alexandroupoli, Greece; pstirop@med.duth.gr; 3Laboratory of Hygiene and Environmental Protection, Medical School, Democritus University of Thrace, 68100 Alexandroupoli, Greece; enena@med.duth.gr; 4Paediatric Respiratory Unit, Paediatric Department, University of Crete, 71500 Heraklion, Greece

**Keywords:** asthma, children, biologic agents, monoclonal antibodies, severe asthma

## Abstract

Undeniably, childhood asthma is a multifactorial and heterogeneous chronic condition widespread in children. Its management, especially of the severe form refractory to standard therapy remains challenging. Over the past decades, the development of biologic agents and their subsequent approval has provided an advanced and very promising treatment alternative, eventually directing toward a successful precision medicine approach. The application of currently approved add-on treatments for severe asthma in children, namely omalizumab, mepolizumab, benralizumab, dupilumab, and tezepelumab have been shown to be effective in terms of asthma control and exacerbation rate. However, to date, information is still lacking regarding its long-term use. As a result, data are frequently extrapolated from adult studies. Thus, the selection of the appropriate biologic agent, the potential predictors of good asthma response, and the long-term outcome in the pediatric population are still to be further investigated. The aim of the present study was to provide an overview of the current status of the latest evidence about all licensed monoclonal antibodies (mAbs) that have emerged and been applied to the field of asthma management. The innovative future targets are also briefly discussed.

## 1. Introduction

Over the last few decades, there has been a shift in the approach of describing the complex and heterogeneous nature of asthma, which can possibly permit the delivery of precision medicine to pediatric patients. Recent advances have classifiedasthma into phenotypes and endotypes in a commendable attempt to group individuals based on common features, pathophysiology and treatment approaches [1,2,3].

In light of the multiple shapes of asthma, which present as a diverse entity with multiple underlying mechanisms, patient responses to treatment vary tremendously. Probably the most common and distinctly identified phenotype of the disease is characterized by increased T-helper 2 (TH2) cytokines and mediators, known as allergic or TH2-high asthma [2,3]. Accordingly, TH2-high asthma is defined by the underlying feature of elevated levels of type 2 inflammation in the airways compared with healthy controls [4]. As asthma endotypes are further elucidated and better understanding of the role of the TH2 pattern of inflammation has been attained, novel biologic therapies targeting these particular pathways have emerged [2]. To date, the available biologic agents were add-ons targeted at individuals with TH2-high severe asthma, but recently a new add-on treatment was approved for patients with no TH2-high asthma [5,6].

Addressing severe asthma refractory to the basic therapeutic toolkit demands examining more advanced, innovative and steroid-sparing agents such as biologics [7,8]. Pediatric severe asthma accounts for a limited percentage of children with asthma, but it also counts for an extraordinarily excessive load of resource utilization and morbidity, even fatal reactions. Severe asthma is conventionally defined as asthma that is uncontrolled despite appropriately prescribed medication and treatment of related issues or that aggravates when high dose treatment is reduced [6,9]. Airflow obstruction-inducing persistent symptoms, needing higher levels of controller therapy is the hallmark [10]. Its diagnosis involves a careful final assessment to exclude potent masquerading disorders accompanied by a differentiation between difficult-to-treat versus severe therapy resistant asthma (STRA) [7,8]. STRA is asthma that has persistently poor control even with the application of maximum standard therapies and control-based management that take modifiable risk factors into account. Patients diagnosed with STRA most often need add-on therapeutic strategies [11]. Aggressive medication management and adherence, the remediation of environmental-inciting inhalants, and treatment of comorbid conditions is obligatory [12]. All the above result in the cost-effective, rational use of biologics which can target certain TH2 mediators, revolutionizing the management of severe asthma.

Biological-based therapeutics are a type of treatment that has been laboratory-produced from living organisms rather than from chemical procedures [2]. All licensed biologic therapies for the treatment of STRA target specific key cytokines and are humanized monoclonal antibodies (mAbs) and a full human mAb, dubbed dupilumab; consequently, the words biologics and mAbs are frequently used synonymously [2]. MAbs originate from a B-lymphocyte clone and bind to a matching epitope. Mammalian cells are their main hosts since they can accurately implement post-translational changes [12,13]. From a biopharmaceutical perspective, the superbly targeted selectivity results in lower toxicity due to optimal binding affinity, leadingmAbs to be regarded as groundbreaking and safe treatments [14].

During the past decade, outstanding progress has been made in the domain of allergic diseases, principally regarding the pathogenetic role of TH2 inflammation. The aim of this study was to review the characteristics of mAbs available for children suffering from STRA to summarize present knowledge in the context of personalized severe asthma management and discuss the future perspective for pediatric and adolescent populations.

## 2. Synopsis of TH2 Dominant Asthma Pathways

TH2-high asthma is characterized by allergic sensitization and eosinophilic inflammation of the respiratory tract, guided by type 2 prototypical cytokines comprising IL-4, IL-5 and IL-13 [12,15]. TH2 inflammation has presented in a group of patients with severe asthma and has been mostly concentrated in serum IgE levels, eosinophilia and exhaled nitric oxide (FeNO), which are biomarkers useful for the selection of the most suitable biologic factor [6,10].

Chronic airway inflammation in asthma is activated by many complicated immunologic pathways. The pathways activation is linked to the exposure to stimulants, such as aeroallergens and viruses or environmental pollutants, which come in contact with the antigen-presenting cells, specifically dendritic cells [2]. Thereafter, dendritic cells trigger naïve T helper cells, switching them into TH1, TH17 and TH2 cells [2]. TH2 cells commence the type 2 pathogenic cascade through signals that cause the production of type 2 cytokines [2,16]. Cytokine expression aids the activation of effector cells that extend airway inflammation and hyperreactivity [16].

TH2 cells secrete IL-4 and mediate IgE class-switching in B cells [4]. Additionally, TH2 cells migrate to the airway epithelium and secrete IL-5 and IL-13 [2,4]. IL-4 and IL-13 have been observed to boost IgE production [16]. IgE coexisting with various allergens prompts the release of acute-acting mediators, such as histamine, leukotrienes, and prostaglandins which act as instigators of vascular permeability and bronchoconstrictor stimuli [2]. Moreover, IL-5 supports maturation and migration of eosinophils [16]. The upregulation of cytokines creates a vicious cycle as advances activate type 2 mediator cells that enhance the immunological response [16].

In depth understanding of the molecular, cellular and biological mechanisms underlying TH2 predominant pathways is constantly expanding. Recent research in severe asthma involve multicenter programs in Europe and in the United States, named the Unbiased Biomarkers for the Prediction of Respiratory Disease Outcome (U-BIOPRED) and the Severe Asthma Research Program (SARP), respectively These programs are aimed at gaining more insights about severe asthma and uncovering better treatments. Corticosteroids have been the mainstay therapy to suppress persistent type 2 inflammation when taken regularly and correctly, although in some cases, they are ineffective, and the asthma is considered refractory. Recently, biologic agents targeting certain proteins related to asthma’s inflammatory routes have been approved, all of which are administered to treat individuals with TH-2 high asthma.

## 3. Approved Biologic Agents

The majority of pediatric patients can achieve good asthma control. However, a minority will have severe asthma even after optimal therapy [6,17]. Initially, inhaled corticosteroids (ICS) brought immense relief from clinical symptoms and lung function indexes [6]. Thus, whereas the majority of children with asthma can be handled in a primary healthcare setting, a few, regardless of good inhaler technique and adherence to step 4 treatment instructions, will need a referral to experts, phenotyping, and add-on treatment [6,18]. Precision or personalized therapy through mAbs has become a possible life-changing choice in the last decade.

There are currently five mAb therapies available for children with severe asthma younger than 18. Not long ago, only omalizumab had approval, originally in the United States since 2003 and subsequently in Europe in 2005, but recently, the European Medicines Agency (EMA) approved mepolizumab and benralizumab, in 2015 and 2017, respectively, dupilumab in 2019 for children [2,17,19], and tezepelumab first in the USA in December 2021 and at the Global Initiative for Asthma (GINA) in 2022 [6,20]. The main characteristics of each of the immunomodulatory mAbs approved for pediatric severe asthma have been reported in Table 1. Table 2 summarizes outcomes confirmed in clinical trials.

### 3.1. Anti-IgE Therapy (Omalizumab)

Omalizumab is an anti-IgE humanized mAb produced using recombinant DNA technology. It is approved for children older than 6 years and focused against the constant region of heavy ε chain of free systemic IgE ([5,39] Table 1). Omalizumab-IgE complex prevents binding FcεR1 on B cells, monocytes, mast cells and basophils, resulting in diminution of the downstream inflammatory cascade, eventually resulting in an attenuated allergic response [5,39]. It was the first approved mAb for a pediatric indication and the treatment of STRA. Presently, it is the only biologic approved for children with inadequately controlled severe persistent asthma under the age of 12 years [6].

Omalizumab is administered via subcutaneous injection every 2 to 4 weeks and has been added to step five since the 2017 GINA guidelines. The dosage (mg) and dosing frequency for asthma is guided by two charts, one for adults and adolescents and the another for children 6 to 12 years old, based on body weight (kg) and serum total IgE levels (IU/mL) prior to the treatment initiation, and ranges from 75 to 375 mg [40]. It is worth noting that those two charts are insufficient in calculating the optimal dosage in some categories of patients with higher baseline IgE and body weight data [40]. Additionally, serum total IgE levels cannot be used as a marker for dose determination as they may raise in the course of treatment and persist for up to a year after treatment cessation [40].

The previously described sequence of molecular events justifies the effectiveness of Omalizumab in clinical practice. Even though there are numerous randomized controlled trials in adults indicating the beneficial outcomes of omalizumab as an alternative to asthma standard controlled therapy, in children the number of clinical trials is still limited. In a detailed review in a total of 25 clinical trials [23], only 3 were exclusively pediatric studies [30,41,42]. Among pediatric studies, 2 included children aged 6 to 12 years [41,42], and one concerns patients aged 6 to 20 years.

Studies of school-aged children and youths on omalizumab in long-term trials have exhibited hopeful effects. Noteworthy, in a placebo-controlled study, omalizumab reduced ICS intake and asthma exacerbations during the steroid-reduction phase and showed improved quality of life [42]. Similarly, another pediatric study demonstrated a reduction in asthma exacerbations by 43% [41]. Additionally, omalizumab nearly extinguishes the autumn/spring peaks regarding exacerbations and lowers the need for controller therapy [30]. Interestingly, in a study of a subset of patients treated with omalizumab, researchers, after isolating peripheral blood mononuclear and dendritic cells and stimulating ex vivo with rhinovirus, found that the decreased expression of the high affinity FCεRI was associated with increased secretion of interferon α (INF-α) from dendritic cells [43]. Thus, the increased IFN-α response to viruses might be one of the underlying mechanisms that explains the reduction of viral-induced exacerbations in fall [43].

Impressively, a 4-year follow-up placebo–control study of children with moderate and severe uncontrolled asthma showed the probability that omalizumab can change the natural course of asthma, as participants were free of ICS or rescue treatment requirement during 3 years after discontinuation of omalizumab [44].

Broadly, in real life, omalizumab is well tolerated. Specifically, approximately 0.2% of patients present anaphylaxis. Local side effects are more common ([6,23] Table 1). Although, initial data suggested a possible causal relationship with malignant tumors, an increase in the risk was not shown [45].

### 3.2. Anti-IL-5 Therapies (Mepolizumad, Benralizumab)

IL-5 is a cardinal cytokine which regulates miscellaneous features of respiratory track eosinophilic inflammation, which is commonly related to augmented asthma severity, including the promotion of eosinophil’s maturation, activation and survival [16,46]. It is well known that eosinophils as well as their chemical mediators have a major role in airway inflammation. Mepolizumab is a murine humanized mAb targeting circulating IL-5 and blocking IL-5/IL-5Rα interaction approved by the Food and Drug Administration (FDA) and EMA as an add-on treatment for children with severe eosinophilic asthma ([12] Table 1). Benralizumab is also a humanized mAb targeting the alpha subunit of the cellular receptor of IL-5, approved by FDA and EMA for severe eosinophilic asthma [12]. Furthermore, it attracts natural killer cells to provoke cellular death of eosinophils [47].

The recommended dose for mepolizumab administration is 100 mg every 4 weeks subcutaneously, and for benralizumab it is 30 mg subcutaneously every 4 weeks. for the first 3 doses. Afterwards, it is 30 mg every 8 weeks [6].

Regarding mepolizumab, two important randomized, double-blind studies, named DREAM and MENSA, were conducted [48,49]. All participants were patients older than 12 with severe asthma and signs of eosinophilic inflammation. Both studies aimed to establish efficacy and safety in response to mepolizumab [48,49]. The researchers demonstrated a remarkable clinical advancement in terms of the rate of asthma exacerbations, resulting in reduction of emergency department visits and hospitalizations. In terms of an increased baseline, it forced the expiratory volume in the first second (FEV1) by approximately 0.1 L compared with the placebo groups [48,49]. In another randomized, double-blind trial, called the SIRIUS study, the authors aimed to determine the effect of mepolizumab on the use of oral glucocorticoids (OCS) in patients older than 16 years with severe eosinophilic asthma [32]. Mepolizumab showed an important glucocorticoid-sparing effect, as the median percentage reduction in OCS was found to be 50% in the mepolizumab group. It also showed a decrease in the rate of exacerbations per year and improved control of symptomatology [32].

Considering benralizumab, two randomized, double-blind, multicenter studies, the SIROCCO and CALIMA study, enrolled patients over 12 who weighed more than 40 kg and aimed to examine the efficacy and safety of benralizumab as an additional option for severe eosinophilic asthma [50,51]. The authors indicated a significant decrement of annual asthma exacerbation rate, a significantly improved pulmonary function, specifically pre-bronchodilator FEV1, and an improvement of the symptoms in those with blood eosinophils more than 300 cells/mL [50,51].

Overall, the studies confirmed that mepolizumab and benralizumab are both effective, safe and well-tolerated treatments for patients with severe asthma and elevated eosinophil count. The most common adverse events were injection-site reactions; anaphylaxis is rare (Table 1).

### 3.3. Anti-IL-4R Therapy (Dupilumab)

IL-4 and IL-13 are two narrowly correlated drivers of TH2 inflammation [52]. IL-4, and to a lesser degree IL-13, animate IgE synthesis; while IL-13 has a fundamental role in airway hyperresponsiveness, mucus production and airway remodeling [16,39]. Dupilumab is a fully humanized anti-IL-4 receptor α mAb, which selectively binds to IL-4Rα, thus inhibiting both IL-4 and IL-13 signaling [36,53]. It is approved as an add-on therapy in adults, adolescents and children older than 6 years, for type 2 severe asthma characterized by requirement for maintenance OCS or increased eosinophils and/or FeNO levels [6].

Dupilumab is administered via subcutaneous injection in patients with severe eosinophilic asthma in a dose of 200 mg or 300 mg every 2 weeks, and 300 mg for OCS-dependent patients or individuals with coexistent moderate/severe atopic dermatitis [6].

In an international, double-blind, phase 3 trial, the QUEST trial, the authors aimed to evaluate its efficacy and safety in randomized selected participants, aged 12 and older, with uncontrolled type 2 OCS-dependent asthma [34]. In this study, patients who received dupilumab showed a significantly lower rate of severe exacerbations compared to the placebo group, improved lung function, as shown by increased FEV1 by approximately 0.32 L, and better asthma control [34]. More pronounced benefits were seen in the subgroup of patients with higher baseline eosinophil count and greater baseline FeNO level [34].

In another international, double-blind phase 3 trial, the VENTURE trial, randomly selected participants older than 12 with OCS-dependent severe asthma were treated with dupilumab to assess its effectiveness in restricting OCS administration [36]. Severe exacerbation rate was lowered by 59%, and the percentage change of OCS was significantly higher in the dupilumab group compared to the placebo group. Furthermore, an improvement of lung function, as FEV1 was by 0.22 L higher, was shown [36].

Ultimately, in another double-blind, placebo-controlled, phase 3 trial, the VOYAGE trial, randomly selected school age children 6 to 11 years old with uncontrolled moderate-to-severe asthma were treated with add-on dupilumab [37]. Among participants, those who received dupilumab had fewer annual severe exacerbations, better lung function in terms of baseline FEV1 and statistically significant greater asthma control compared to the placebo group.

Regarding the safety profile of dupilumab, it is a well-tolerated mAb. The most frequently described adverse events are injection-site reactions and transient blood eosinophilia (Table 1).

### 3.4. Anti-TSLP Therapy (Tezepelumab)

Tezepelumab is a humanized IgG2λ mAb that inhibits thymic stromal lymphopoietin (TSLP), an airway epithelial cytokine [12,20], and it has shown a wider therapeutic action in both type 2 high and low severe asthma patients [38]. Therefore, it is the first approved biologic agent for severe asthma with no restrictions regarding phenotype or existence of a certain biomarker for ages older than 12. TSLP is crucial for TH2 immunity, which is involved in asthma pathophysiology, as it induces the expression of TH2-associated inflammation drivers, such as IL-4 and IL-5 [54].

It is administered subcutaneously, in a dose of 210 mg every 4 weeks [6], as this therapeutic scheme is linked with approximately 90% of maximum effect regarding FeNO levels and asthma exacerbation rates [55].

In a double-blind, phase 2 study, the PATHWAY trial, in adults with uncontrolled moderate-to-severe asthma, those who received tezepelumab benefited independently if they had TH2-high or TH2-low inflammation [56]. A multicenter, randomized, double-blind phase 3 trial, the NAVIGATOR trial, of patients aged 12 to 80 years with severe, uncontrolled asthma, those who received tezepelumab had fewer exacerbations and better lung function, asthma control and health-related quality of life than those who received a placebo [57].

In another double-blind, placebo-controlled, phase 2 trial, from 27 medical centers in several countries, the CASCADE trial, in randomly selected adults aged 18 to 75 with moderate-to-severe asthma, individuals treated with tezepelumab displayed a higher reduction in airway submucosal eosinophils, irrespective of baseline blood eosinophil levels [58]. In addition, the tezepelumab administration group showed a reduced airway hyperresponsiveness to mannitol [58]. Lastly, in another double-blind, placebo-controlled trial, the UPSTREAM trial, reduced airway hyperresponsiveness and eosinophil levels in bronchoalveolar lavage and airway tissue was shown in adults in the tezepelumab group [59].

Regarding the safety profile of tezepelumab, it is a well-tolerated mAb as the expressed adverse events are similar with the placebo groups [6]. The most frequently described adverse events are injection-site reactions, pharyngitis, arthralgia and back pain (Table 1).

### 3.5. Selecting a Biologic Therapy in Children Younger than 18 Years

GINA recently published a pocket guide for difficult-to-treat asthma [60]. Relevant to the present review is the consideration of potential predictors of good asthma response among anti-IgE, anti-IL-5, anti-IL-4R and anti-TSLP therapy, if locally accessible and cost-effective. Assuredly, it is never a simple, one-dimensional decision when choosing among available biologic agents.

In particular, whilst raised IgE is one of the most apparent indicators for treatment with omalizumab, it cannot estimate the possibility of a good response. Similarly, asthma exacerbations were decreased with either high or low blood eosinophils or with high or low FeNO [6,61], although allergen-driven symptoms and childhood onset of asthma are related to probable improvement of treatment with omalizumab [6]. Considering anti-IL-5 therapies, in addition to an improved benefit among those with higher peripheral blood eosinophils, individuals with more frequent exacerbations, adult-onset asthma and those with nasal polyps are more likely to benefit according to the GINA guidelines [2,6]. Additionally, dupilumab is expected to have beneficial effect in patients with higher blood eosinophils and FeNO [34]; however, it is not suggested if blood eosinophils are higher than 1500/µL [6,37]. Dupilumab is currently approved in children older than 6 years who are not on maintenance OCS, and it can be used in moderate-to-severe atopic dermatitis and chronic rhinosinusitis with nasal polyposis [6,62]. While newly approved for patients older than 12 years, tezepelumab may be administrated in patients with no phenotype, regardless of allergic status [20]. However, it showed superior clinical advancement when blood eosinophils and/or FeNO are higher, but there is insufficient evidence in individuals on maintenance OCS [6,60].

Based on current treatment knowledge, it seems that children older than 6 with elevated IgE levels should be treated with omalizumab, while those with an increased eosinophils number (>300 μL) could be treated with mepolizumab or dupilumab if they are 6 to 12 years old or with mepolizumab, dupilumab, benralizumab or tezepelumab if they are older than 12 [6]. Of course, it is worth noting that tezepelumab can be administered even with no elevated TH2 markers [6].

Determining when to abort therapy is as much of a challenge as figuring out the appropriate candidate for receiving biologic asthma therapy. The GINA guidelines suggest considering a switch into a different biologic if there is no significant response, recommending 4 months at least to evaluate treatment response [60]. Additionally, even if a good response seems to be attained, patients should be re-evaluated every 3 to 6 months, and OCS should be gradually reduced or stopped once adequate control has been achieved [6]. Biologic therapies should be used as add-on therapies; thus patients should be maintained on at least a moderate dose of ICS [6].

## 4. Future Targeted Therapies

Today, dozens of mAbs are used in medical clinical practice, and others are being examined in clinical trials, introducing an innovative type of management for diverse non-malignant diseases, including asthma. The progress in genetic and biomedical fields, along with a more holistic approach of pathophysiological directions of pediatric asthma, have identified new targets. There is no doubt that the growing availability of biologic agents to treat children with severe asthma gives another alternative to improve their overall quality of live.

Reslizumab is an mAb that has a mechanism of action similar to mepolizumab, as it binds to circulating IL-5 with high specificity and disrupts eosinophil maturation, assisting cell death [6,63]. It was approved for use in 2016 in adults with severe, uncontrolled, type 2 asthma, particularly in those with high blood eosinophils [63,64]. From two multicenter, double-blind, phase 3 trials which enrolled patients aged above 12, reslizumab has been reported to decrease the exacerbation rate and extend the time to exacerbation compared to placebo groups [63]. Another phase 3 trial, with participants aged 12 to 75 years with blood eosinophil count above 400 cells/mL, has shown an increase in FEV1 values compared with baseline values, and this outcome was dose-dependent [65]. Finally, a retrospective study examining the real-world effects in adults with severe eosinophilic asthma on add-on treatment with reslizumab showed improved clinical and patient-reported outcomes and remarkable declines in associated healthcare resource use [66].

The FDA Adverse Event Reporting System (FAERS) displayed a good safety profile respecting reslizumab in children, with only one mild adverse event, eosinophilic esophagitis, and only one severe adverse effect, chronic cholecystitis. Nevertheless, although a number of advantages were observed, there are still concerns about the intravenous administration as subcutaneous administration was not found to be effective in adults [67]. The wider use of this medicine and further randomized controlled trials are required to evaluate its clinical effectiveness and safety in children [12].

The epithelial cytokine IL-33 which is released by airway epithelial cells in response to stimulants, such as allergens, is implicated in asthma susceptibility and severity. Clinical trials have demonstrated the efficacy of itepekimab an anti-IL-33 mAb, and astegolimab, an IL-33 receptor inhibitor, in adult asthma outcomes [38]. Research of anti-IL-33 mAb is in its infancy. Further future clinical trials involving a wider spectrum of patients with severe asthma, for example children and patients with TH2-low asthma, are needed.

Add-on treatments are essential to alter the natural history of respiratory disease and restrain lung function decline. Mechanistic research has succeeded in introducing various biologic agents that target specific pathways of type 2 inflammation. Indisputably, the utilization into clinical practice of approved mAbs and future add-on treatments that are in the pipeline is more than fascinating.

## 5. Future Perspective/Future Directions

There is a preponderance of studies implemented in adults compared to children, with the exception of trials regarding omalizumab. Impending observational and experimental studies focusing on pediatric populations will be valuable as we introduce more mAbs to treat children and adolescents. Additionally, follow-up studies are mandatory to certify long-term safety and to define the adequate treatment period of add-on treatments to designate response and total duration of treatment to determine optimal effectiveness. Moreover, additional research is urgently needed to assess novel, noninvasive, prognostic response biomarkers that will facilitate the selection of the appropriate agent based on each patient asthma endotype. Likewise, to date, more high-quality randomized controlled trials are demanded to define the differences between the licensed mAbs to match the most appropriate tailored add-on treatment for an individual when standard care is not adequate.

In brief, the rise of the new class of add-on treatments, called biologic agents, has led us to a new epoch in severe asthma management. Data are not sufficient enough with regard to the efficacy and safety of biologic agents in the pediatric population, and even more in certain subpopulations, including minority children and children from low-income countries. However, setting up national registries of known cases of severe, uncontrolled childhood asthma would be one way to ensure that the needs of these patients are being met. 

## 6. Conclusions

Future medications can enhance the quality of life of children with severe asthma and diminish the disease burden as they reach their adult years by modifying the natural history of asthma with personalized precision care. Unequivocally, the relatively recent delineation of distinct immune phenotypes and, even more recently, endotypes, have led to a new era in pediatric asthma, the era of biologic agents. However, we still have some unanswered questions such as the optimal duration of treatment with biologics in children, the discovery of novel noninvasive prognostic biomarkers, and whether the new agents, contrary to corticosteroids, will be able to change the natural course of asthma.

Biologics seem to have a targeted mechanism of action through definite type 2 inflammatory pathways. However, the discovery of a flawless medicine is still a matter of debate. More precise phenotyping and the participation of more children in phase 3 studies of new biological therapies will remain important and continuous challenges.

## Figures and Tables

**Table 1 jpm-12-00999-t001:** Licensed monoclonal antibodies licensed for use in pediatric severe asthma.

mAb	Target	Mechanism of Action	Applicable Population (Age in Years)	Common Eligibility Criteria	Safety Profile
Omalizumab [6,21,22,23]	IgE	Binds to Fc part of free IgE, inhibits IgE from binding to its receptors (FcεR1 receptors), reducing free IgE levels and down-regulating receptor expression, resulting in prevention of activation and releasement of TH2 inflammation mediators	≥6 years	Exacerbations during the last year, and sensitization to inhaled allergens, and total serum IgE levels (generally for IgE-mediated persistent allergic asthma)	Headache, fever (in aged 6 to 12), injection-site reactions (erythema, edema, pain, pruritus), sporadic/very rare cases of anaphylaxis (approximately 0.2%)
Mepolizumab [6,22,24,25]	IL-5	Binds to circulating IL-5 and prevents its interaction with IL-5 receptor α, reducing eosinophil levels	≥6 years	Severe exacerbations during the last year, and baseline blood eosinophils above determined count (>150 cells/μL or >300 cell/μL)	Headache, injection-site reactions, eczema, nasal congestion, rare cases of anaphylaxis
Benralizumab [6,22,25]	IL-5 receptor α	Binds to IL-5 receptor α, inhibiting IL-5 pathway, causing apoptosis of eosinophils	≥12 years	Same as mepolizumab	Headache, injection-site reactions, pharyngitis, rare cases of anaphylaxis
Dupilumab [6,26]	IL-4 receptor α	Binds to IL-4 receptor α, blocking the signaling induced by IL-4 and IL-13 signaling, downregulating TH2 inflammatory pathways	≥6 years	Severe exacerbations during the last year, and existence of type 2 biomarkers, or necessity for maintenance OCS	Injection-site reactions, transient blood eosinophilia, rare cases of vasculitis with eosinophilic granulomatosis with polyangiitis
Tezepelumab [6,20]	TSLP	Binds to circulating TSLP, inhibiting interaction with its receptor and its’ upstream action in asthma inflammatory cascade	≥12 years	Severe exacerbations during the last year, considered in patients with no type 2 biomarkers	Injection-site reactions, pharyngitis, arthralgia, back pain, rare cases of anaphylaxis

Abbreviations: mAb, monoclonal antibody; Ig, immunoglobulin; IL, interleukin; OCS, oral corticosteroids; TSLP, thymic stromal lymphopoietin.

**Table 2 jpm-12-00999-t002:** Clinical Outcomes: comparison among monoclonal antibodies.

mAb	Quality of Life	Exacerbations	Asthma Control	Lung Function Parameters	Blood Eosinophil Number	Median OCS Use
Omalizumab [6,16,27,28,29,30]	Improved	Decreased	Improved	No significant difference	No significant difference	Decreased
Mepolizumab [31,32]	Improved	Decreased	Improved	Improved	Decreased	Decreased
Benralizumab [31,33]	Improved	Decreased	Improved	Improved	Decreased *	Decreased
Dupilumab [34,35,36,37]	Improved	Decreased	Improved	Improved	Decreased **	Decreased
Tezepelumab [6,20,38]	Improved	Decreased	Improved	Improved	Decreased ***	No significant difference

Abbreviations: mAb, monoclonal antibody; OCS, oral corticosteroids. * Regarding anti-IL-5 therapies: mepolizumab, reslizumab (for adults), and benralizumab, all distinctly decreased blood eosinophil levels, however, benralizumab caused almost total depletion. ** Dupilumab was related with transient eosinophilia in some patients, which returned to baseline levels by the end of treatment session. *** Extrapolating from adult clinical trials.

## Data Availability

Not applicable.

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
