# Peer review of "Biologic Therapies in Pediatric Asthma"

_jpm, 2022, doi:10.3390/jpm12060999_

Round 1

Reviewer 1 Report

In the paper the authors are reviewed  the current status and the latest evidence of biologic therapy in pediatric asthma.

The paper is divided into 6 main chapters; Introduction, Synopsis of TH2 Dominant Asthma Pathways, Approved Biologic Agents,  Future Targeted Therapies,  Future Perspective/Future Direction, and Conslusion.

Finally, the authors rightly open questions related to the application of biological therapy in asthma such as the optimal duration of therapy, the discovery of novel non-invasive biomarkers and whether the new biologic agents will be able to change the natural course of  childhood asthma.

The paper is well written with a key message; all licensed monoclonal antibodies been applied to the field of asthma management in children.

Author Response

Thank you for your comments. We made some minor English editing and we modified our manuscript in accordance with GINA's 2022 Guide, which was released after the time of our first submission. Additionally, we added some of the latest references.

Reviewer 2 Report

I have read with interest the article titled:Biologic therapies in pediatric asthma“ which has been submitted to the Journal of Personalized Medicine. The aim of the present study was to provide an overview of the current status of the latest evidence about all licensed monoclonal antibodies that have emerged and been applied to the field of asthma management. The topic selection is highly actual.

The paper is quite well-written but unfortunately did not include the latest insights or references. Its contents are of great interest. However, a review and clarification of several points appear needed. The scientific quality of the manuscript is quite good, but many errors are present. The article needs to be revised with newer available medications and guidelines. I would consider adding some more references pointing out some recent advances in the field.

I suggest inserting new knowledge into the text according to GINA's 2022 Guide to Severe Asthma as well as the latest references such as Brusselle GG, Koppelman GH. Biologic Therapies for Severe Asthma. N Engl J Med. 2022 Jan 13;386(2):157-171. 

Accordingly, enter new drugs in Tables 1 and 2 (e.g. Tezepelumab (SC), Reslizumab (IV)).

line 228.. older than 12 years..

Dupilumab now also approved for children ≥6 years with severe eosinophilic/Type 2 asthma, not on maintenance OCS (Bacharier, NEJMed 2021)

After line 246 …insert VOYAGE Trial results (reference 36)

Section 3.4. Selecting a biologic therapy in children younger than 18 years needs to be completely revised (according to the GINA severe asthma guide in 2022)

Author Response

Reviewer 2: I suggest inserting new knowledge into the text according to GINA's 2022 Guide to Severe Asthma as well as the latest references such as Brusselle GG, Koppelman GH. Biologic Therapies for Severe Asthma. N Engl J Med. 2022 Jan 13;386(2):157-171.

Respond: Thank you for your comment, by the time of submission (April 29) GINA 2022 wasn't released. Consequently, we made all the appropriate changes regarding latest instructions/guidelines (e.g. changes regarding step 5 options in 6 to 11 years old group). We also added a few more of the latest references, among them Brusselle, G.G.; Koppelman, G.H. Review. Biologic Therapies for Severe Asthma. N. Engl. J. Med. 2022, 386, 157-171, as suggested.

Reviewer 2: Accordingly, enter new drugs in Tables 1 and 2 (e.g. Tezepelumab (SC), Reslizumab (IV)).

Respond: Thank you for your comment. Regarding Tezepelumab, it has been moved from future therapies section to the section of Approved Biologic Agents in accordance with your comments and GINA's 2022 Guide. Additionally, we added some new references, in order to shed light into recent advances, such as:

Diver, S.; Khalfaoui, L., Emson, C.; et al;. Effect of tezepelumab on airway inflammatory cells, remodelling, and hyperresponsiveness in patients with moderate-to severe uncontrolled asthma (CASCADE): a double-blind, randomised, placebo-controlled, phase 2 trial. Lancet Respir. Med. 2021, 9, 1299-1312.

Sverrild, A.; Hansen, S.; Hvidtfeldt, M.;et al. The effect of tezepelumab on airway hyperresponsiveness to mannitol in asthma (UPSTREAM). Eur. Respir. J. 2021, 59, 2101296.

Hoy, S.M. Tezepelumab: First Approval. Drugs. 2022, 82, 461-468.

Puzzovio, P.G.; Eliashar, R.; Levi-Schaffer, F. Tezepelumab administration in moderate-to-severe uncontrolled asthma: Is it all about eosinophils? J. Allergy Clin. Immunol. 2022, 149, 1582-1584.

Ly, N.; Zheng, Y.; Griffiths, J.M.; et al. Pharmacokinetic and pharmacody-namics modeling of tezepelumab to guide phase 3 dose selection for patients with severe asthma. J. Clin. Pharmacol. 2021, 61, 901-912.

Regarding Reslizumab, as it is approved for those ≥ 18 years, and our aim as to give the current status of mAbs in pediatric population, we did not add it in our tables, rather we have mentioned it and provided information in future therapies section. Furthermore, we have added some new information and two new references.

Wechsler, M.E.; Peters, S.P.; Hill, T.D.; et al. Clinical outcomes and health-care resource use associated with reslizumab treatment in adults with severe eosinophilic asthma in real-world practice. Chest. 2021, 159, 1734-1746.

Bernstein, J.A.; Virchow, J.C.; Murphy, K.; et al. Effect of fixed-dose subcutaneous reslizumab on asthma exacerbations in patients with severe uncontrolled asthma and corticosteroid sparing in patients with oral corticosteroid-dependent asthma: results from two phase 3, randomised, double-blind, placebo-controlled trials. Lancet Respir. Med. 2020, 8, 461-474.

Reviewer 2: line 228.. older than 12 years..

Respond: Thank you for your comment. You are absolutely right, we corrected it, in line with your comment and GINA's 2022 Guide.

Reviewer 2: Dupilumab now also approved for children ≥6 years with severe eosinophilic/Type 2 asthma, not on maintenance OCS (Bacharier, NEJMed 2021)

Respond: Thank you for your comment. You are absolutely right, we corrected it.

Reviewer 2: After line 246 …insert VOYAGE Trial results (reference 36)

Respond: Thank you for your comment. We have added the important results of ref.36

Reviewer 2: Section 3.4. Selecting a biologic therapy in children younger than 18 years needs to be completely revised (according to the GINA severe asthma guide in 2022)

Respond: Thank you for your comment, you are absolutely right. We made the necessary alterations, as the recent changes in GINA brought new aspects in this evolving field.

Round 2

Reviewer 2 Report

The authors modified the manuscript in accordance with GINA's 2022 Guide to Severe Asthma and added some of the latest references. I suggest accepting the paper in its current form.

Author Response

Once again, on behalf of all authors, I would like to thank you very much for your time.